# Oxidative Damage to Mitochondria Enhanced by Ionising Radiation and Gold Nanoparticles in Cancer Cells

**DOI:** 10.3390/ijms23136887

**Published:** 2022-06-21

**Authors:** Farnaz Tabatabaie, Rick Franich, Bryce Feltis, Moshi Geso

**Affiliations:** 1School of Sciences, RMIT University, Melbourne, VIC 3000, Australia; farnaz.taba89@gmail.com; 2School of Health & Biomedical Sciences, RMIT University, Bundoora, VIC 3083, Australia; brycenf@gmail.com

**Keywords:** gold nanoparticles (AuNP), radiation, mitochondria, cancer cell lines, reactive oxygen species

## Abstract

Gold nanoparticles (AuNP) can increase the efficacy of radiation therapy by sensitising tumor cells to radiation damage. When used in combination with radiation, AuNPs enhance the rate of cell killing; hence, they may be of great value in radiotherapy. This study assessed the effects of radiation and AuNPs on mitochondrial reactive oxygen species (ROS) generation in cancer cells as an adjunct therapeutic target in addition to the DNA of the cell. Mitochondria are considered one of the primary sources of cellular ROS. High levels of ROS can result in an intracellular state of oxidative stress, leading to permanent cell damage. In this study, human melanoma and prostate cancer cell lines, with and without AuNPs, were irradiated with 6-Megavolt X-rays at doses of 0–8 Gy. Indicators of mitochondrial stress were quantified using two techniques, and were found to be significantly increased by the inclusion of AuNPs in both cell lines. Radiobiological damage to mitochondria was quantified via increased ROS activity. The ROS production by mitochondria in cells was enhanced by the inclusion of AuNPs, peaking at ~4 Gy and then decreasing at higher doses. This increased mitochondrial stress may lead to more effectively kill of AuNP-treated cells, further enhancing the applicability of functionally-guided nanoparticles.

## 1. Introduction

Exposing cells to radiation results in oxidising events that change atomic structure due to the direct interactions of the radiation with the target’s macromolecules or through products of water radiolysis [1]. Petkau 1987 showed the first biochemical alterations, which happen throughout or shortly after exposure to the radiation, and were believed to be accountable for most of the consequences of irradiation in mammalian cells. However, oxidative changes can still continue for days and months after the first exposure, likely due to the constant generation of reactive oxygen (ROS) and nitrogen (RNS) species [2].

Mitochondria are inherited in most cells, typically numbering in the hundreds per cell, and are multifunctional. They are known to store and harvest energy released via oxidative phosphorylation. These organelles have a small amount of DNA which encodes rRNAs and tRNAs as well as proteins for respiration. Mitochondria interact via membrane contact, signal transduction, and vesicle transport, in order to regulate biosynthesis, energy metabolism, immune responses, and cell turnover. Therefore, they are considered critical intracellular junctions for vital interactions with cells’ other organelles. Thus, cells do not use enough energy when mitochondria are damaged, and unused oxygen and fuel molecules build up and cause damage [3]. It is also known that most cancer cells retain mitochondrial functions, including respiration and energy harvesting [4,5]. Mitochondria also modulate ROS generation and redox status, and initiate apoptosis by activating mitochondrial protein tyrosine phosphatases (mtPTP) [6]. Mitochondria are essential in creating and developing phenotype cancers [1].

Since the seminal research of Hainfeld et al. and others, we have investigated the roles of various nanoparticles in the radiosensitisation of cells in the radiotherapy context [7,8]. Of the wide-ranging studies that described the biological effects of AuNPs, several reported elevated levels of reactive oxygen species for AuNPs of differing size, shape, and surface functionalization [9]. By combining gold, which has a high atomic number and perceived nanostructure, with the ability of sub-100nanometre objects to penetrate the tumor vasculature, it was proven that AuNPs acted effectively to increase the dose to tumor sites. We have also explored individual effects of nanoparticles on cells, including in the absence of radiation, e.g., cell mobility [10] and enhancement of radiation effects by gold nanoparticles for superficial radiation therapy [8]. Observations show that (i) nanoparticles have poor penetration into the cell nucleus—which is the location of most DNA, the primary target for radiotherapy—coupled with (ii) the short-ranged dose enhancement around NPs by Auger electrons. These observations lead us to explore the role of mitochondrial damage. Quantitative analysis of NP-enhanced radiation damage and stress to mitochondria may lead to an exploitable targeting opportunity that is independent of the NP’s ability to penetrate the nucleus. Few reports have demonstrated a role for ROS or the involvement of mitochondria as a mechanism of AuNP radiosensitisation [11]. Taggart et al. conducted similar research to us, with the main difference being that they explored the effect of ionising radiation on mitochondria in the kV range [12]. In this study, cells were exposed to doses in the megavoltage range, which is the therapeutic range. Moreover, we quantified results using two different methods.

Our main objective was to determine and quantify the combined effects of ionising radiation (IR) and AuNPs on the mitochondria of cells (Figure 1). This study characterized the use of MitoSOX in flow cytometric detection of mitochondrial superoxide in mitochondrial cells, with and without ionising radiation and AuNPs. MitoSOX and flow cytometry are used to quantify the superoxide produced by mitochondria of cells that are exposed to ionising radiation with and without AuNPs. The outcomes are compared and validated for both of these conditions. The results of the MitoSOX assay were also compared with those from a Seahorse XF Analyser in order to determine the reliability of the assays in detecting the relative changes in mitochondrial superoxide formation under megavoltage irradiation of various doses. Such beams represent the predominant therapeutic energy range used. We proved that the Mitochondria could be affected by irradiation and gold nanoparticles. However, this is dose dependent. The effect of radiation and AuNPS increases up to 4 Gy and by increasing the dose, it drops down. This could indicate that beyond 4 Gy dose and for this type of cells other modes of cell death are dominating.

## 2. Results

### 2.1. Cytotoxicity of AuNPs

The MM418-C1 and DU145 cell lines were treated with biocompatible 1.9-nanometre AuNPs in solution at concentrations ranging from 0 to 1.5 mM for 24, 48, and 72 h, in order to determine the reactions and tolerance levels of the treated cell lines to the AuNPs in the absence of radiation. Cell viability was measured using an MTS assay and was reported as a ratio of the optical absorbance of the AuNP-treated groups to control groups (with no AuNPs) at the same time point [13]. As shown in Figure 2, the addition of AuNPs to cells did not significantly impact the viability of cells: the AuNPs are not toxic to either cell line at all tested concentrations. The DU145 cells exhibit the typical response visible after 48 h, followed by stimulated recovery at 48–72 h. In Figure 3, we see that the response of both cell lines is similar, as shown for the 48-hour time point. The NP concentration selected for the remainder of the study was 1 mM.

### 2.2. Cell Survival Assay

The survival fraction of cells after exposure to a range of radiation doses from 0 Gy (control) to 8 Gy, with and without AuNPs, was measured by performing a clonogenic assay [14]. Cell survival curves are plotted in Figure 4, and dose enhancement factors (DEF) are determined based on Equation 2. DU145 cells exhibited a DEF of 1.17, while MM418-C1 was slightly more sensitive at DEF = 1.21. Results confirm that the inclusion of AuNPs enhances the net radiological effects of IR on both cell lines (6) by about 17% to 21% for these cells, respectively.

### 2.3. Effect of Radiation and AuNPs on Mitochondria

#### 2.3.1. Flow Cytometry Technique

Overgeneration of ROS during mitochondrial respiration may cause DNA damage. MitoSOX red was used to detect ROS levels in order to determine mitochondrial damage during exposure to IR. MitoSOX red is a mitochondrial superoxide indicator that can be oxidized by superoxide in mitochondria to produce red fluorescence [15]. In order to address the effect of IR and AuNPs on mitochondria of cancer cells (DU145 and MM418-C1), we measured acute mitochondrial superoxide formation using flow cytometry (Figure 5 and Figure 6). Figure 5 shows that ROS production in DU145 cell lines peaked at approximately 4 Gy, before apparently decreasing toward 6 Gy. For the second cell line tested (MM418-C1) we also applied up to 8 Gy in order to confirm this decrease in mitochondrial ROS production at higher doses; that trend was observed to continue.

#### 2.3.2. Extracellular Flux (XF) Analysis

The Seahorse XF Analyser, cell mitochondrial stress test determines cellular bioenergetics following mitochondrial metabolic stress. Oxygen consumption rate (OCR) measurements were taken in real-time during a day for both cell lines at basal levels and following sequential addition of mitochondrial respiration inhibitors oligomycin, carbonyl cyanide-p-trifluoromethoxyphenylhydrazone (FCCP), and a combination of antimycin A and rotenone.

Basal respiration (Figure 4) represents the energy demands of cells under basal conditions; the oxygen consumption of basal respiration is used to meet ATP synthesis, and results in mitochondrial proton leak. The oxygen consumption rate was calculated at every measurement point for each of the 20 wells of the XF24 plate used in the mitochondrial stress test. Every well-produced bioenergetic trace (Figure 7) gives a mean value for each technical replicate within a group. Basal respiration was stable before adding modulating drugs, which, once injected into the extracellular media, could successfully isolate the desired components of OXPHOS, in which electron transport-linked phosphorylation or terminal oxidation is the metabolic pathway, as expected. In Figure 7, the OCR increases with radiation dose to a maximum at 4 Gy, before decreasing at higher doses. In all cases, the addition of AuNPs raised the OCR, with the observed effect being greater for the DU145 cells than for the MM418-C1 cells, although the general trend and peak at 4 Gy were consistent for both.

As a compensatory response, an increase in glycolysis is frequently observed. Mitochondria are required for cell energy metabolism, and play an important role in cell death. Mitochondrial failure can be caused by changes in mitochondrial respiration or the balance of pro-apoptotic and anti-apoptotic proteins [16]. During the same measurement period, the change in proton concentration was also measured against time in order to determine the extracellular acidification rate (ECAR), predominantly a measure of lactic acid formed during aerobic glycolysis (Figure 8). A low and stable ECAR value was observed during the basal measurement period for the same experiment. Upon introducing oligomycin, a rapid increase in ECAR was observed, reflective of increased glycolytic metabolism due to the inhibition of ATP synthase. We can see similar results to OCAR in the ECAR calculation as well. Like OCR, ECAR increases with dose to a maximum at or near 4 Gy before decreasing at higher doses of 6 and 8 Gy. The results for the MM418-C1 cells showed a much stronger ECAR effect than that for DU145, indicating that MM418-C1 clearly switched to glycolysis.

## 3. Discussion

In this study, we investigated the contribution of mitochondria to oxidative stress and radiation effects on two different types of cancer cells when they were exposed to radiation in combination with AuNPs. The main aim of this study was to quantify the effects of the ionising radiation on a target that is outside the cell’s nucleus, particularly the mitochondria. DNA is considered to be the only important target in investigations of the radiation-induced cell killing [17]. The reason for this is because other cell structures and organelles, if damaged, can be reproduced by the DNA. However, mitochondria are not produced by nuclear DNA, and have a level of redundancy because there are many in the cells. Mitochondria are also known to initiate and trigger cell death via apoptosis. Hence, damage to mitochondria could lead to the triggering or inhibiting of this process. For these reasons, mitochondria are often viewed as secondary targets to nuclear DNA for their importance in cell death via radiation. Many studies have addressed the effects of IR on the nucleus of the cell [7,18], and in particular on nuclear DNA, which is conventionally considered the primary target of radiotherapy. Other mechanisms are known to contribute to cell death during radiotherapy, and it is well known that mitochondria are vital for cell viability. However, studies about radiation-induced changes in the mitochondria are lacking [19]. It therefore may be desirable to develop a deeper and more quantitative understanding of radiation-induced damage to mitochondria, as this may lead to opportunities that exploit mitochondrial contributions to cell death. Quantifying the enhancement effects of nanoparticles may facilitate new avenues toward the targeted use of NPs—especially given that NPs are known to have poor penetration into the nucleus [20]. Mitochondria are considered one of the primary sources of intracellular reactive oxygen species (ROS). The overgeneration of ROS can induce a state of intracellular oxidative stress that can lead to permanent cell damage. Quantitative analysis of ROS can thus be used to measure cellular injury [21]. A variety of clinical disorders have been shown to include “mitochondrial dysfunction,” which usually coincides with the occurrence of excessive ROS production and oxidative stress. A known effect of oxidative stress is damage to, and mutation of, mitochondrial DNA [22]. Current experimental data imply that a rise in the tumoricidal efficacy of radiation therapy may be achievable by targeting mitochondria [23]. Mitochondrial damage could result in cell death either by direct disruption of its essential functions, or because mitochondria are the initiator of programmed cell death, or apoptosis.

To our knowledge, the only research based on investigating the effects of IR and NPs on mitochondria is documented by Taggart et al. (2014) [12]. This study determined the role of mitochondrial function in AuNP radiosensitisation. Their study was based on exposure to low-energy X-rays of the kilovoltage (kV) range, where the strongly energy-dependent photoelectric effect combines with the high atomic number of the NPs to physically enhance the radiation energy deposited. However, low energy X-rays are rarely used as a treatment source in contemporary radiotherapy. Our study is based on the response to megavoltage X-rays, which are of a much higher energy, achieving greater penetrability and hence, deeper and higher dose deposition with a skin-sparing effect. Most, if not all, current radiotherapy techniques are based on megavoltage beams, very similar to the type used in this research. For photons at these high energies, the increase in interaction probability of high-atomic-number metallic nanoparticles is negligible; yet despite this, radio-enhancement has been observed [12]. The increased radiosensitisation is attributed to two primary influences: (i) the inhibition of nuclear-DNA repair, which may dominate over any increase in actual DNA damage [24], and (ii) the hyper-localised secondary radiation field in the near-region of the individual nanoparticles [25]. Various studies have confirmed that there is almost no toxicity at low concentrations of AuNPs [8,14]. We first validated the optimal concentration of NPs to be used in our study by determining cytotoxicity levels in both types of cells. Our results confirmed that up to a concentration of about 1.5 mM, AuNPs did not cause cell death in either cell line as shown in Figure 2 and Figure 3. The cell survival curves (Figure 4) show that the presence of AuNPs in both cancer cell lines exposed to 6-megavolt X-ray beams generated by a Linac caused radiosensitisation of the cells. This type of radiation is most commonly used in treating cancer patients via radiotherapy. The outcome of these studies also shows that these DE values are only significant at low-energy radiation on the order of kV. However, these low-energy X-rays are not commonly used in RT treatments. Therefore, it is likely that NPs can be used to preferentially target organelles in the cytoplasm where the majority of NPs penetrate. In our study, indicators of mitochondrial stress are quantified using two techniques which were different to those used by Taggart et al. [12]. The role of NPs as radiation dose enhancers has been documented in the literature and in previous studies by our group [8]. The consistent outcome of much research has shown that these particles enhance radiation doses significantly at low-energy X-rays [in the kV range]. In contrast, NPs have almost no effects in the MV range of high-energy X-rays. It has been shown that the inclusion of gold NPs in cells before irradiation enhances cell death, which is termed dose enhancement DE. This DE inflicted by NPs has been exclusively attributed to cell nuclear DNA damage [26].

Various studies have confirmed that there is almost no cytotoxicity at low concentrations of AuNPs [27]. We first validated the optimal concentration of NPs to be used in our study by determining toxicity levels in both types of cells. Our results of cytotoxicity analysis confirmed that up to a concentration of about 1.5 mM, AuNPs did not have a cytotoxic effect on either cell line, as shown in Figure 2 and Figure 3. The cell survival curves (Figure 4) show that the presence of AuNPs in both cancer cell lines exposed to 6-megavolt X-ray beams generated by a Linac causes cells’ radiosensitisation. This is a type of radiation most commonly used in treating cancer patients via radiotherapy. The outcome of these studies shows also that these DE values are only significant at low-energy radiations on the order of kV. However, these low-energy X-rays are not commonly used in RT treatments. Our results show that DE levels (represented by DEF values) in prostate and melanoma cancer cells indicated notably higher DEFs of 1.08 and 1.09, respectively, for high-energy [megavoltage] beams which are most frequently used in radiotherapy treatments.

Dose enhancement is usually caused by secondary electrons generated by the interaction of X-ray photons with atomic matter, and the production of ROS through their interactions with water molecules. These ROS and secondary electron “free radicals” could have an impact on various structures and organelles in the cells. We explored these effects using flow cytometry with the MitoSOX fluorophore as explained in the Materials and Methods section. MitoSOX is one of the most commonly used probes for detecting cellular ROS as expressed by the cell’s mitochondria related to external stimuli. Fluorophores accumulate in mitochondria due to their positive charge and act as an ROS level indicator [28]. It should be noted that MitoSOX will detect ROS that are generated by mitochondria as a stress response from exposure to IR and NPs (20).

These data derived from the two different cancer cell lines show that when these cells are exposed to ionising radiation, the level of ROS increases up to doses of 4 Gy; beyond 4 Gy, the level decreases (Figure 5 and Figure 6). With the inclusion of AuNPs, the ROS level produced from mitochondria was significantly higher than the radiation-only group; thus, we can conclude that AuNPs enhanced the effect of radiation on the mitochondria of the cells. In order to validate these results, we also used the Seahorse system to measure the effect of radiation on the mitochondria of both cell types.

The pattern of OCR in the two types of cancer cells in response to the ionising radiation-induced and NP-induced stress was measured using the MitoStress Kit, which directly measures the activity of electron transport chains in mitochondria. Most of the reactions involved in cellular respiration occur in the mitochondria. If mitochondria are injured, they generally consume more oxygen [29,30]; taking more oxygen generates more ROS. As seen in Figure 7, part a, ionising radiation caused a significant increase in the OCR levels at a dose of 4 Gy. Similarly in MM418-C1 cells, the highest OCR is at 4-Gy; and once the dose is increasing, OCR decreases.

In this study, we determined basal respiration. The tricarboxylic acid cycle (TCA cycle) requires the electron transport chain (ETC) in the mitochondria, uses oxygen as a terminal electron acceptor, and is coupled to ATP production by oxidative phosphorylation. It can completely oxidise the major nutrient substrates glucose, glutamine, and fatty acids, into CO_2_ and H_2_O. Carbonic anhydrase can catalyse the conversion of CO_2_ into bicarbonate and protons [31]. We identified the types of exogenous nutrient substrates that can be oxidized, and the rates at which they can be oxidized under the experimental conditions used. Our results show that maximal respiration (Figure 7) can be a good approximation of the stress response and of the total number of mitochondria in a cell. The increase in the level of mitochondrial respiration is a clear indicator of stress. Under conditions of high ROS, the cellular response could be to increase the number of mitochondria in order to cope with the physiological stress; this increased number of mitochondria would generate more ROS. In summary, both ionising radiation and AuNPs had a significant effect on cell function in both cell lines. At the organelle level, mitochondria, as the main sites of oxidative metabolism, are almost certain to be involved. Since ROS can be mediators of the ionising radiation-induced apoptosis [32], mechanisms that regulate ROS levels, such as antioxidant enzymes, may be critical in protecting cells from ionising radiation-induced cell death. Our results show that the effect of radiation and AuNps on mitochondria increases up to 4 Gy, and at higher doses, it decreases steeply. This increased mitochondrion stress may lead to more effectively kill in AuNP treated cells, further enhancing the applicability of functionally-guided nanoparticles.

## 4. Materials and Methods

In this study, different methods were used to investigate the effect of IR and the inclusion of AuNPs on the mitochondria of two types of cancer cells. Firstly, we investigated what the optimal dosage of AuNPs on our cells would be, in order to ensure that it was not toxic to our cells. Then, we exposed cells to ionising radiation, with and without AuNPs, in order to plot the cell survival curve. Eventually, we measured the effect of IR and AuNPs on mitochondria by using different methods, including flow cytometry (FCM), in order to detect, identify, and count specific cells. Seahorse XF technology measures the flux of oxygen, the oxygen consumption rate (OCR), the flux of protons, and the extracellular acidification rate (ECAR).

### 4.1. AuNP Preparation

A biocompatible solution of round AuNPs was acquired from Nanoprobes Inc. (Yaphank, NY, USA). This solution contains AuNPs with a core diameter of about 1.9 nm, stabilized with a highly water-soluble organic skeleton [33]. AuNPs were diluted in media with 10% FBS and 1% penicillin. Full media without AuNPs were run as a control. The AuNP solution was diluted using a cell culture medium to create the final concentration of 0.5 mM or 0.0985 mg/mL.

### 4.2. Cell Culture

In this study, human primary melanoma (MM418-C1) cells (QIMR, Brisbane, Australia) and DU145 prostate cancer cells (human epithelial cells ATCC^®^ HTB-81™) were used. MM418-C1 were cultured and maintained in RPMI 1640 (Gibco^®^), 10% FBS, and 1% penicillin-streptomycin. DU145 cells were cultured and maintained in MEM Alpha + GlutaMAX™ and 15-millimolar HEPES (Thermo Fisher Scientific, Waltham, MA, USA) and 10% FBS (Gibco^®^), and 1% antibiotics (penicillin-streptomycin; Gibco^®^). The cell lines were initially cultured and grown to about 80% confluence in a 75 cm^3^ flask, and then were subcultured in a 1:3 ratio with trypsin EDTA (Gibco^®^). The incubation conditions during the experiments were 37 °C with 5% CO_2_ in a humidified environment.

### 4.3. AuNPs Cytotoxicity Assay

In order to assess the toxicity of AuNPs on our cell lines, cell viability was measured using the CellTiter 96 Aqueous One Solution cell proliferation assay kit MTS assay (Promega Corporation, Fitchburg, WI, USA). MM418-C1 and DU145 cell lines were seeded in 96-well plates with 5 × 10^3^ cells per well, and incubated at 37 °C with 5% CO_2_ in a humidified environment. Cells were incubated for 24 h and then treated with various concentrations of AuNPs (44): 0.0 (control), 0.5 mM, and 1.5 mM. The cell viability was analysed at 24, 48, and 72 h after the inclusion of AuNPs. Cytotoxicity assays were used to indicate the probability of toxic effects of AuNPs on cells without any exposure to radiation [34].

At times of 24, 48 and 72 h after treating with AuNPs, the medium was removed and 100 µL of cultured medium was supplied with 10 µL of CellTiter 96^®^ Aqueous One solution. Cell proliferation assay was added to the cells. Immediately after adding MTS, the optical absorbance of the formazan was measured at 490 nm using a CLARIOstar^®^ microplate reader (BMG LABTECH, Mornington, Australia) to determine the background (BG). Then, the plates were incubated for just one hour, after which the optical absorbance was measured. Cell survival fractions are expressed as percentages relative to the control groups, as shown in Equation (1).
(1)$$\mathrm{Viability}\;\%=\frac{\mathrm{Absorbance}\;\mathrm{of}\;\mathrm{control}\;\mathrm{cells}\;-\;\mathrm{BG}}{\mathrm{Absorbance}\;\mathrm{of}\;\mathrm{cells}\;\mathrm{with}\;\mathrm{AuNPs}\;-\;\mathrm{BG}}\;\times \;100$$

### 4.4. Cell Irradiation

The cells were irradiated with 6-megavolt X-ray beams generated by a medical linear accelerator (LINAC) (Elektra Synergy, Stockholm, Sweden) located at the Australian Radiation Protection and Nuclear Safety Agency (ARPANSA), Yallambie, Australia. The radiation was delivered as a single fraction for each dose, i.e., 0 Gy for control groups, and from 2 to 8 Gy for treated groups. The given dose was measured from calculations using an ion chamber before cell irradiation. All experimental plates were uniformly exposed to IR; since the cell lines used for this study are adherent, a monolayer of cancer cells forms at the bottom of the 6-well plates. A water-equivalent layer with a thickness of 4 cm placed on top of the plates provided sufficient backscatter radiation to achieve as much electric equilibrium as possible, as shown in Figure 9.

### 4.5. Cell Survival Assay

Clonogenic assay was the technique we used for plotting the cell survival curves [14]. According to the radiation dose, various numbers of cells (Table 1) were seeded in 6-well plates, and cells were incubated for 24 h at 37 °C with 5% CO_2_ in a humidified environment. Cells were treated with 0.5 mM of AuNPs, and were then incubated for 24 h. We renewed the culture medium before irradiation was administered. Cells were incubated for 14 days in order to produce colonies containing about 50 cells. After this incubation time, cell colonies were fixed in (3:1) methanol/acetic acid for 5 min and stained with 0.5% crystal violet. The stained cell colonies were gently rinsed with water and allowed to dry for 24 h (Figure 10). The number of colonies was manually counted using ImageJ software [35].

The results are expressed as ratios relative to the control groups. The dose enhancement factor (DEF) was calculated at D20 using Equation (2), where D20 (control) is the required dose to decrease 20% of the viability of irradiated cells compared to non-irradiated cells in the control group. D20 (AuNPs) represents the dose that reduces 0% of the viability of irradiated cells compared to non-irradiated cells in groups treated with ANPs.
(2)$$E=\frac{D20\left(\mathrm{control}\right)}{D20\left(\mathrm{GNPs}\right)}$$

### 4.6. Measurement of Mitochondrial ROS

Cells were seeded in 6-well plates (2 × 10^6^ cells per mL). Then, after 48 h, cells were treated with 1 mM AuNP and different doses of IR for another 24 h, 48 h, or 72 h, followed by incubation with 5 μM of MitoSOXTM at 37 °C for 30 min in the incubator. The MitoSOXTM signal was visualized using flow cytometry. MitoSOXTM (Thermo Fisher, Waltham, MA, USA) was used to measure mitochondrial ROS formation. In summary, logarithmic growth phase cells were taken in order to make cell suspensions. Then, the 0.3-millilitre cell suspension containing 1.3 × 10^5^ cells was selected and fused with 0.1 mL of 20-micromolar MitoSOXTM. After 0.5 h of incubation, compounds of different concentrations (final volume: 1 mL) were added, and the fluorescence was continuously monitored for 20 min using flow cytometry (BD Biosciences FACSJazz, Pleasanton, CA, USA).

### 4.7. Measurement of Mitochondria OCR and ECAR

In this section, the oxygen consumption rate (OCR) and extracellular acidification rate (ECAR) of mitochondria were measured using Seahorse XF24 equipment (Seahorse Bioscience Inc., North Billerica, MA, USA) (4). In brief, cells were seeded at 120,000 cells per well and treated with AuNPs and IR for 24 h in a 1-millilitre medium. Before the measurements, plates were equilibrated in a CO_2_-free incubator at 37 °C for 1 h. The XF assay protocol was set to measure the change in dissolved oxygen concentration from within a fixed 7-microlitre volume of extracellular media (50) over a 4-minute time. The resulting metric is expressed as picomoles per minute (pmol/min). Five consecutive measurement points were made in order to establish a continuous basal respiration period before the first drug, oligomycin (1 μM), was injected into the extracellular media to inhibit ATP synthase. Phenylhydrazone (FCCP) (2 µM) was then injected to uncouple the proton gradient across the inner mitochondrial membrane, and antimycin (0.5 μM) was added to inhibit complex III, and consequently the electron transport chain (ETC). Absolute OCR measurements are recorded at 8-min intervals. Each interval begins with 2 min of agitation to allow the extracellular media to mix, followed by 2 min of waiting time and then 4 min of measurement upon the transient fixed-volume micro-chamber formation, yielding a mix/wait/measure setting of 2/2/4 min. Each drug injection was followed by two subsequent OCR measurements [36].

### 4.8. Analysis

All data presented in this article are presented as averages of at least three independent experiments. The two groups’ statistical comparison was made using an unpaired *t*-test using IBM SPSS Statistics 25 (IBM Corporation, Armonk, NY, USA). Results are presented as mean ± SEM.

## 5. Conclusions

Mitochondria are responsible for programmed cell death, so if it is stressed, it can lead to cell death. Many of the reactions involved in cellular respiration happen in the mitochondria. We demonstrated the induction of ROS production in two different cancer cells in response to the ionizing radiation-induced stress and AuNps in combination. Mitochondria are affected by irradiation and gold nanoparticles in combination which causes malfunctioning mitochondria and eventually leads to cell death. Results indicate that the effect of radiations and AuNps increases up to 4 Gy in mitochondria, and by increasing the dose beyond this value, it drops down. We can conclude around 4 Gy is the optimal dose for cell damage through mitochondria and higher than 4Gy cell death happens in different ways. Following other references, we believe up to 4 Gy, the ratio of mitochondrial damage is rising, but after 4 Gy, other organelles play a role in cell damage, and the effect of mitochondria is reduced. This conclusion can back up with other references. However, further investigation into this matter is beyond the objective of this study.

## Figures and Tables

**Figure 1 ijms-23-06887-f001:**
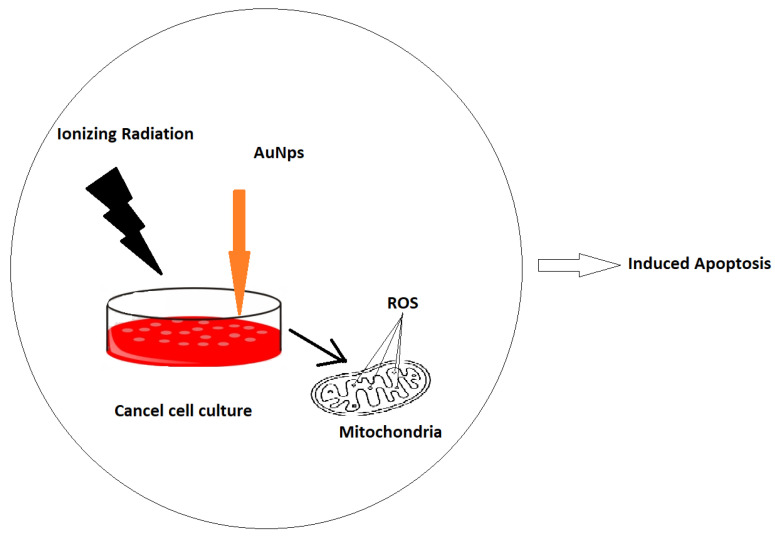
Illustration of cellular mitochondria exposed to ionising radiation in the presence of nanoparticles that can amplify the pathway of ROS-induced cell death.

**Figure 2 ijms-23-06887-f002:**
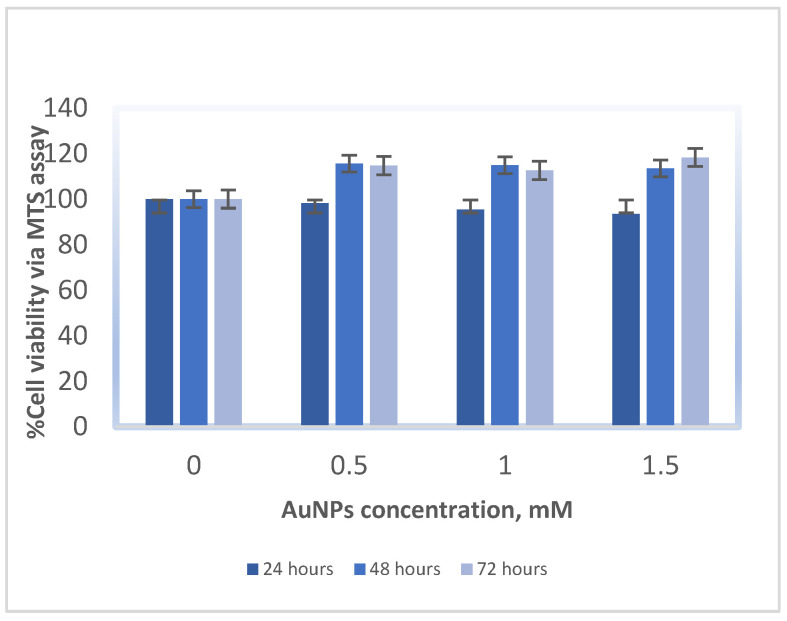
Cell viability (MTS assay) for DU145 cell line 24, 48, and 72 h after being treated with various concentrations of AuNPs. Data are represented as mean values ± SD of three replicate samples.

**Figure 3 ijms-23-06887-f003:**
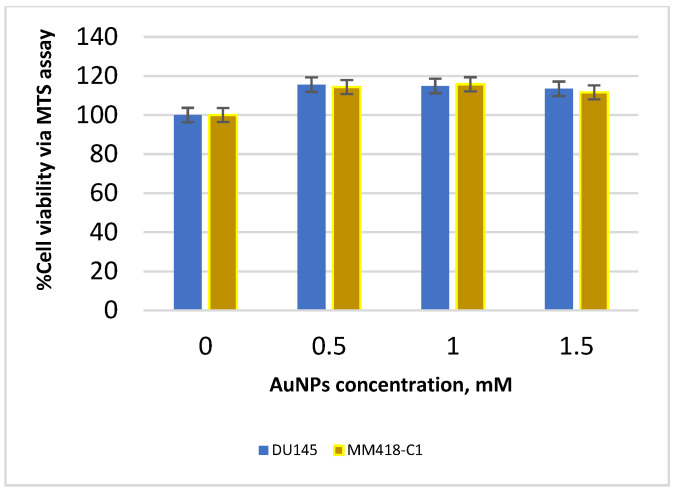
Comparison of cell viability (MTS assay) for DU145 and MM418-C1 cell lines 48 h after being treated with various concentrations of AuNPs. Data are represented as mean values ± SD of three replicate samples.

**Figure 4 ijms-23-06887-f004:**
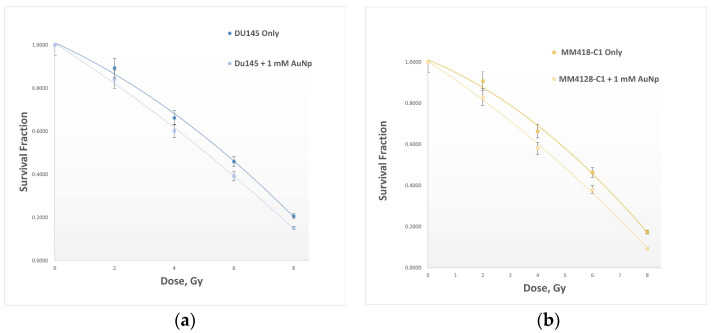
Cell survival curves quantifying the net 6-megavolt radiation dose enhancement caused by the impact of AuNPs on two cancer cell lines: (**a**) DU145, and (**b**) MM418-C1, both with and without AuNPs. Data are represented as mean values ± SD of three replicate samples.

**Figure 5 ijms-23-06887-f005:**
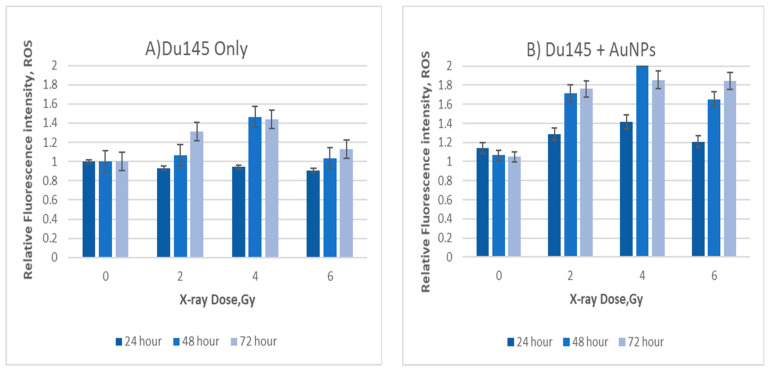
Relative mitochondrial ROS levels in DU145 cells after irradiation (**A**) without AuNPs and (**B**) with AuNPs. Cells were exposed to doses from 0 to 6 Gy. Data are represented as mean values ± SD of three replicate samples.

**Figure 6 ijms-23-06887-f006:**
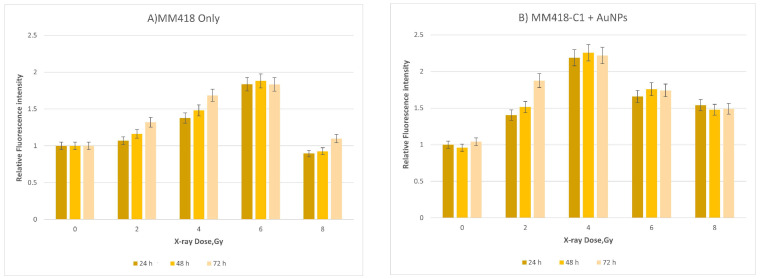
Relative mitochondrial ROS levels in MM418-C1 cells after irradiation (**A**) without AuNPs and (**B**) with AuNPs. Cells were exposed to doses from 0 to 8 Gy. Data are represented as mean values ± SD of three replicate samples.

**Figure 7 ijms-23-06887-f007:**
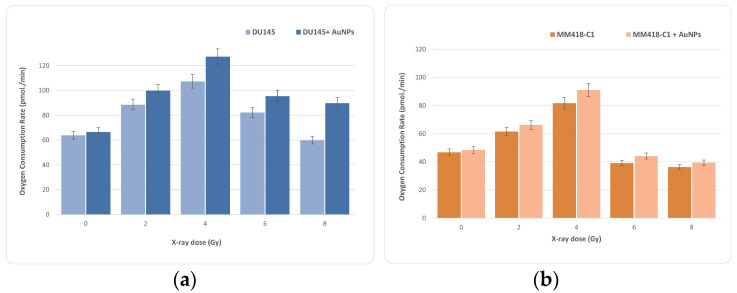
Oxygen consumption rate from a mitochondrial stress test of irradiated DU145 (**a**) and MM418-C1 (**b**) cell lines. Data represent mean values from 20 wells ± SD of three replicate samples.

**Figure 8 ijms-23-06887-f008:**
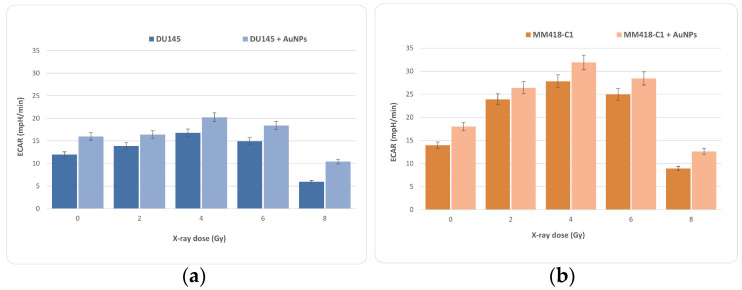
Data from the glycolysis stress test which represent extracellular acidification rate (ECAR) traced from a mitochondrial stress test when DU145 (**a**) and MM418-C1 (**b**) cell lines were irradiated with and without AuNPs. Data represent mean values from 20 wells ± standard deviation of three replicate samples.

**Figure 9 ijms-23-06887-f009:**
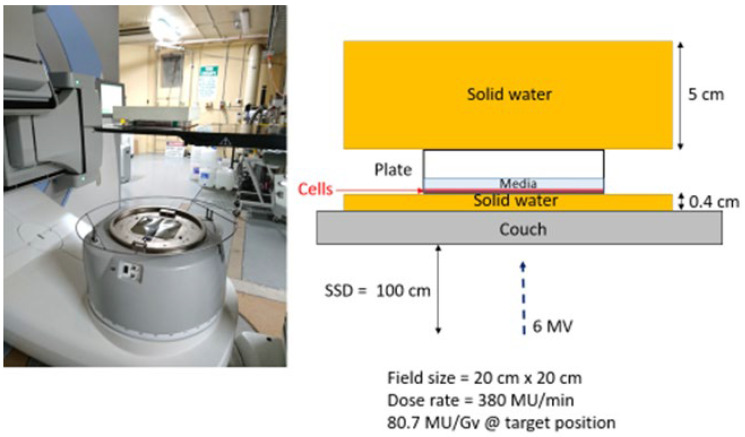
Cell irradiation setup performed at ARPANSA.

**Figure 10 ijms-23-06887-f010:**
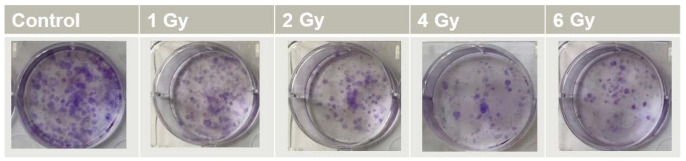
Clonogenic assay performed in 6-well plates, with clones produced by DU145 prostate cancer cells exposed to different doses of IR.

**Table 1 ijms-23-06887-t001:** Cell counts of DU145 and MM418-C1 cancer cells per well which were seeded in 6-well plates in order to perform a clonogenic assay.

Dose (Gy)	Number of Cells per Well
0	500
1	1500
2	2000
4	2500
6	3000

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
