# Peer review of "Oxidative Damage to Mitochondria Enhanced by Ionising Radiation and Gold Nanoparticles in Cancer Cells"

_ijms, 2022, doi:10.3390/ijms23136887_

Round 1
Reviewer 1 Report
The article is in serious need of improvement. There are absolutely no statistical comparisons of the data. The work lacks details of the preparation and characterization of nanoparticles. Without the above revision, I see no reason to review the article in detail.
1) MitoSOX is shown to be distriburted in mitochondria but also in cytosol. Consider this the image of MitoSOX with mitochondrial tracker with fluorescent microscopy.
2) At a minimum, you need to use another dye, in addition to MitoSox. Possibly DCF-DA
3) Since the journal is serious and highly rated, additional methods should be used to confirm the results of apoptosis induction. At least PCR. This may allow the authors to get closer to the molecular mechanism of enhancing the effects of radiotherapy using nanoparticles.
4) The legends for the figures are not very informative. Information is needed on the number of repeats, the number of cells in each plate etc.
5) Figure 1 should be significantly more informative and reflect the proposed mechanism of apoptosis induction in accordance with the results of the authors. In the presented form, it does not carry a semantic load for a professional reader.
6) Why do authors capitalize Mitochondria?
7) The quality and resolution of the drawings should be improved.
8) Literature references should be in [].
9) In general, the work is framed carelessly. For example ThermoFisher, Massachusetts, America. Maybe USA? Or South America? And there is a lot of this kind of negligence. The article should be carefully and thoroughly revised by the authors.
Author Response
Reviewer 1:
1) MitoSOX is shown to be distriburted in mitochondria but also in cytosol. Consider this
the image of MitoSOX with mitochondrial tracker with fluorescent microscopy.
We chose Mitosox because it indicates the ROS produced from mitochondria. That way, we thought
we could investigate the effect of IR on the mitochondria of cells. This is the reference that gives the
full function of Mitosox;
“ Polster BM, Nicholls DG, Shealinna XG, Roelofs BA. Use of potentiometric fluorophores in the
measurement of mitochondrial reactive oxygen species. methods in enzymology 2014 Jan 1 (Vol.
547, pp. 225-250). Academic Press”
ROS detection by MitoSOX is in principle like ROS detection by hydroethidine. Mito-hydroethidine is
oxidized by superoxide to mito-2-hydroxyethidium and by other ROS to mito-ethidium (Robinson et
al., 2006; Zielonka and Kalyanaraman, 2010). The positively charged oxidation products are retained
by polarized mitochondria and exhibit red fluorescence upon interaction with mitochondrial DNA. In
addition to oxidation by ROS, photo-oxidation of MitoSOX causes formation of mito-ethidium
(Zielonka et al., 2006).
2) At a minimum, you need to use another dye, in addition to MitoSox. Possibly DCF-DA
As we mentioned earlier, the main aim of this work is to evaluate the effect of Ionizing Radiation on
mitochondria. So, we chose Mitosox as one method and the XF analyzer (using Mito stress test kit) as
a second method. Both results perfectly matched. Another Dye we used for mitochondria, was Mito-
tracker. However, that was only for imaging purposes, and that way, we cannot evaluate any effect on
mitochondria. We have used DCF-DA in the past – including for Moshi’s papers. These dyes will still
react with superoxide, though it is true that they are not specific to the mitochondria.
“Youkhana EQ, Feltis B, Blencowe A, Geso M. Titanium dioxide nanoparticles as
radiosensitisers: an in vitro and phantom-based study. International Journal of Medical Sciences.
2017;14(6):602.”
3) Since the journal is serious and highly rated, additional methods should be used to confirm
the results of apoptosis induction. At least PCR. This may allow the authors to get closer to
the molecular mechanism of enhancing the effects of radiotherapy using nanoparticles.
Thank you for the suggestion. Sure, there is lots of other stuff we could have done. But it was outside
the scope of what I was trying to achieve with the paper. Due to time constraints and to have a
cohesive and complete paper focused on the main aims, which are synergistic effects of AuNPs and
IRs on mitochondria. The ratio of apoptosis to other cell death methods has been documented in the
literature. This part of our work was to confirm what has been documented and link it to the effects of
mitochondria. We are currently aiming to run a series of measurements to investigate further and
quantify the level of cell death by ionising radiations via apoptosis compared to other pathways and
combination with NPs in the cells during irradiation. However, including such results in this paper will
swallow it into an extensive document with more than one aim, confusing the readers. Moreover, in
this ratio with IRs, we will only add the role of NPs and their effects on it.
4) The legends for the figures are not very informative. Information is needed on the number of
repeats, the number of cells in each plate etc.
Thanks, we added more information to almost all our figures
5) Figure 1 should be significantly more informative and reflect the proposed mechanism of
apoptosis induction in accordance with the results of the authors.
As you suggested, figure 1 has been updated. The main aim of figure 1 is to simplify this work's whole
idea. I hope it is acceptable now.
6) Why do authors capitalize Mitochondria?
Fair point. All fixed in the main text.
7) The quality and resolution of the drawings should be improved.
Figures are uploaded separately with high resolution.
8) Literature references should be in [].
References fixed.
9) In general, the work is framed carelessly. For example ThermoFisher, Massachusetts,
America. Maybe USA? Or South America? And there is a lot of this kind of negligence. The
article should be carefully and thoroughly revised by the authors.
Thank you for your comment. It has been modified. I hope it is acceptable now.
Reviewer 2 Report
The manuscript "Oxidative Damage to Mitochondria Enhanced by Ionizing Radiation and Gold Nanoparticles in Cancer Cells" is devoted to studying the ability of AuNP to increase the effect of ionizing radiation on cancer cells through mitochondria-related oxidative reactions. The purpose of increasing the efficiency of anticancer therapy is of high importance and the results of the study are interesting, but some of the authors' conclusions are not convincing.
- The manuscript is written carelessly: large parts of the manuscript repeat themselves (for example, the sentences in lines 277-286 completely coincide with those in lines 297-305) , the same words begin with either a capital or a small letter, there are a lot of typos (for example, Fig. 10 is mentioned in the text in lines 186 and 333, but there is not any Fig. 10 in the manuscript).
- The main hypothesis of the authors is that AuNPs enhance production of ROS by mitochondria, which results in decreased cell viability. However, the results in Fig. 5 and Fig. 6 contradict this main hypothesis, because both figures demonstrate much higher production of ROS in cells not exposed to AuNPs compared to those exposed to AuNPs at all irradiation levels.
- The discussion of the results is somewhat confusing. More references are needed to corroborate the authors' hypothesis. For instance, such phrases as "If mitochondria are injured need more oxygen consumption" (lines 332-333) and those in lines 341-343 (see below) should be supported by appropriate references. Moreover, the authors state that "The level of mitochondria respiration increase is a clear indicator of its stress. If you have high ROS, then it could be that cells build up an increased number of mitochondria with 4 Gy to cope with the physiological stress, and an increased number of mitochondria are generating more ROS" (lines 341-343). However, the authors also demonstrate the switching of cells to aerobic glycolysis under the same conditions (Fig. 8). So, it is not clear what substrates are oxidized by mitochondria if there is a concurrent increase in substrate oxidation in glycolysis in the cytoplasm. More literature references are needed to explain the observed effects.
- No other evidence is presented that could confirm the primary role of mitochondria in the observed effects. The increased oxygen consumption rate could be a result of either ROS production or increased aerobic lactic glycolysis that was also demonstrated by the authors. For instance, morphological studies of mitochondria could help.
- The authors hypothesize that at ionizing radiation dose up to 4 Gy, the dominant cell death method could be through apoptosis and, at higher doses, the probability of cell death via other channels such as necrosis dominates. This statement can be easily checked using fluorescent apoptosis/necrosis staining of cells, considering that the authors apparently have an access to a flow cytometer. The paper by Rainaldi et al. is not a very good corroboration to the hypothesis because, in the paper by Rainaldi et al., much higher dose of ionizing radiation (30 Gy) induced necrosis.
- According to Fig. 2 and Fig. 3 all cells treated with AuNP demonstrated higher viability compared to control cells, which was measured by MTS reduction. Gold nanoparticles can have their own influence on the conversions of MTS stain. Was the ability of AuNP to promote MTS reduction to formazan in the absence of cells tested?
- Please provide more information on the coating of AuNP ("organic skeleton" is too uncertain).
- Please provide direct results on cell viability after the treatment with ionizing radiation and AuNP
- The plot in Fig. 4 looks a bit strange, because although the authors mention that the maximal effect of AuNP in promoting cell injury under ionizing radiation treatment is observed at 4 Gy, and at higher doses of radiation the effect of AuNP decreases, the curves do not demonstrate any specific behavior around 4Gy.
- In lines 337-338, the authors wrote "we identified the type of exogenous nutrient substrates that can be oxidized and the rates at which they can be oxidized under the experimental conditions used", but there is no data on nutrient substrates in the manuscript.
Author Response
Reviewer 2
1. The manuscript is written carelessly: large parts of the manuscript repeat
themselves (for example, the sentences in lines 277-286 completely coincide with
those in lines 297-305), the same words begin with either a capital or a small letter,
there are a lot of typos (for example, Fig. 10 is mentioned in the text in lines 186 and
333, but there is not any Fig. 10 in the manuscript).
Thanks for calling our attention to such details. we reviewed the manuscript
thoroughly and corrected these points that the reviewer mentioned.
2. The main hypothesis of the authors is that AuNPs enhance production of ROS by
mitochondria, which results in decreased cell viability. However, the results in Fig. 5
and Fig. 6 contradict this main hypothesis, because both figures demonstrate much
higher production of ROS in cells not exposed to AuNPs compared to those exposed
to AuNPs at all irradiation levels.
There was a mistake in the figures ordering. The correction has been done.
3. The discussion of the results is somewhat confusing. More references are needed to
corroborate the authors' hypothesis. For instance, such phrases as "If mitochondria
are injured need more oxygen consumption" (lines 332-333) and those in lines 341-
343 (see below) should be supported by appropriate references. Moreover, the
authors state that "The level of mitochondria respiration increase is a clear indicator
of its stress. If you have high ROS, then it could be that cells build up an increased
number of mitochondria with 4 Gy to cope with the physiological stress, and an
increased number of mitochondria are generating more ROS" (lines 341-343).
However, the authors also demonstrate the switching of cells to aerobic glycolysis
under the same conditions (Fig. 8). So, it is not clear what substrates are oxidized by
mitochondria if there is a concurrent increase in substrate oxidation in glycolysis in
the cytoplasm. More literature references are needed to explain the observed
effects.
References are added to augment the discussion points.
The paragraph is added to clarify this point.
“As a compensatory response, an increase in glycolysis is frequently observed.
Mitochondria are required for cellular energy metabolism and play an important role
in cell death. Mitochondrial failure can be caused by changes in mitochondrial
respiration or the balance of pro-apoptotic and anti-apoptotic proteins”
4. No other evidence is presented that could confirm the primary role of mitochondria
in the observed effects. The increased oxygen consumption rate could be a result of
either ROS production or increased aerobic lactic glycolysis which was also
demonstrated by the authors. For instance, morphological studies of mitochondria
could help.
That is a very fair point. Such measurements as suggested by the reviewer could
quantify the attribution of the effects of the IRs and AuNPs on ROSs generations in
comparison with other sources but first, this is beyond the scopes of this paper and
second, the exact contribution due to the two external stimuli {AuNPs and IRs} is not
the aim of this study but its relative increase with the increase of radiation dose and
with to without NPs.
5. The authors hypothesize that at ionizing radiation dose up to 4 Gy, the dominant cell
death method could be through apoptosis and, at higher doses, the probability of
cell death via other channels such as necrosis dominates. This statement can be
easily checked using fluorescent apoptosis/necrosis staining of cells, considering
that the authors apparently have an access to a flow cytometer. The paper by
Rainaldi et al. is not a very good corroboration to the hypothesis because, in the
paper by Rainaldi et al., much higher dose of ionizing radiation (30 Gy) induced
necrosis.
Yes, we agree but here quantification of a known phenomenon as the reference given
in this section and many others can be found in the literature is not the chief aim of
this paper, and including data from such measurements will make the paper lengthy
beyond the limits indicated by the journal. However, we are planning to follow up with
these experiments and perhaps make the apoptosis/necrosis discussion a bit more
speculative.
6. According to Fig. 2 and Fig. 3 all cells treated with AuNP demonstrated higher
viability compared to control cells, which was measured by MTS reduction. Gold
nanoparticles can have their own influence on the conversions of MTS stain. Was
the ability of AuNP to promote MTS reduction to formazan in the absence of cells
tested?
This is a good point. We did measure the media only by MTS assay and corrected all
measurements based on that. But, We have not tested AuNps in the MTS assay in the
absence of cells. However, this test has been done by a published paper and confirms
that AuNps has no influence on the conversions of MTS stain. Here is the reference that
we can mention in this regard:
“Kazmi F, Vallis KA, Vellayappan BA, Bandla A, Yukun D, Carlisle R. Megavoltage
radiosensitization of gold nanoparticles on a glioblastoma cancer cell line using a
clinical platform. International Journal of Molecular Sciences. 2020 Jan;21(2):429.”
7. Please provide more information on the coating of AuNP ("organic skeleton" is too
uncertain).
More information is added to the text.
8. Please provide direct results on cell viability after the treatment with ionizing
radiation and AuNP.
This has already been done. Please check figure 4.
9. The plot in Fig. 4 looks a bit strange, because although the authors mention that the
maximal effect of AuNP in promoting cell injury under ionizing radiation treatment
is observed at 4 Gy, and at higher doses of radiation the effect of AuNP decreases,
the curves do not demonstrate any specific behavior around 4Gy.
Please note that figure 4 is the demonstrating cell survival curve which is showing the
effect of Ionizing radiation on whole cells not just in one part of cells. However, later,
with two different methods, we did investigate the effect of IR on the mitochondria of
cells.
10. In lines 337-338, the authors wrote "we identified the type of exogenous nutrient
substrates that can be oxidized and the rates at which they can be oxidized under
the experimental conditions used", but there is no data on nutrient substrates in the
manuscript.
More information is added and further clarified, thanks for your time.
Reviewer 3 Report
The authors have studied the effect of radiation doses on two cancer cell lines in the presence and absence of Au naoparticles, focusing on effects on mitochondrial function. The focus on the effect on mitochondrial function and the outcome for ROS production, cell viability and relative amounts of apoptosis vs necrosis makes this a unique study. The result that the presence of the nanoparticles sensitizes the cells moderately to radiation damage is interesting. Some minor revisions would be:
(1) line 72, what is meant by 'superoxide drive of mitochondria'?
(2) line 73, should be 'with and without AuNPS'
(3) line 97, please clarify 'typical injury response'
(4) line 116, correct the reference
(5) Does it matter if the AuNPS are located inside the cell or nearby and adjacent?
(6) Would there be any way to differentiate the relative magnitude and nature of the effects of radiation on the nucleus from that on mitochondria
Author Response
Reviewer 3
(1) line 72, what is meant by 'superoxide drive of mitochondria'?
Mitochondrial superoxide production is an important source of reactive oxygen species in
cells.
(2) line 73, should be 'with and without AuNPS'
Thank you for bringing this to our attention. It has been edited in the text.
(3) line 97, please clarify 'typical injury response'
It has been modified in the text.
(4) line 116, correct the reference
Done
(5) Does it matter if the AuNPS are located inside the cell or nearby and adjacent?
This is always of interest in NP studies. Based on published papers, AuNps are
distributed randomly inside cells.; however, this can depend on cell shapes and size.
(6) Would there be any way to differentiate the relative magnitude and nature of the
effects of radiation on the nucleus from that on mitochondria
This is a very interesting question. This would be beyond the scope of this work and
needs more in-depth biology work. Although this can be something to think about.
Round 2
Reviewer 1 Report
I was not entirely satisfied with the answers to the key remarks. Additional research methods are critically needed for this work.
My decision regarding the article has not changed
Author Response
Dear Reviewer,
We agree with your suggestion of an extra experiment will improve the paper, unfortunately, that would make it longer and the results are highly unlikely to change the conclusion of this research. We believe that the effect of the ionizing radiation on mitochondria has not been investigated enough compared to ample studies of their effects on the cell's Nucleus. In this work, we chose two different & reliable methods to quantify these effects. Moreover, this line of study is gaining momentum and as evidence to that many very recent articles are targeting this particular point of investigation. [example: Surface Functionalization of Organosilica Nanoparticles With Au Nanoparticles Inhibits Cell Proliferation and Induces Cell Death in 4T1 Mouse Mammary Tumor Cells for DNA and Mitochondrial-Synergized Damage in Radiotherapy by Chihiro Mochizuki published in Frontiers in Chemistry May 2022”. However, the two reviewers believe that the methods presented in this research are insufficient proof for the mitochondria’s damage by IRs and AuNPs. Although, we still think our work has a great value, and the two methods we chose are unique and reliable and the data are obtained from many measurements. We have confidence in our paper because two different approaches resulted in similar outcomes and that confirms our findings and conclusions.
Further works are underway for establishing quantitative and certain conclusions levels of these findings. These experiments will focus on another way “3rd” of determination of the levels of the oxidative mitochondria damages and levels of apoptosis to the necrosis enhanced by the gold nanoparticles
Again, we highly appreciate your efforts in helping making our paper better
Warm Regards,
Associate professor Moshi Geso
Reviewer 2 Report
Unfortunately, I am not satisfied with the authors' response. The manuscript is called "Oxidative Damage to Mitochondria...", but the damage to mitochondria themselves was not demonstrated. As I mentioned before, more data are needed to show the changes in mitochondria (for example, microscopy studies) and the primary role of mitochondra in the observed effects. The phenomenon of AuNP promoting the death of irradiated cells is very interesting but the participation of mitochondria must be more significantly substantiated
Author Response
Dear Reviewer,
Thank you for your time and valuable comments.
Please note that, The seahorse base experiment is the device that measures oxidative damage to mitochondria; we used the kit that is named “Mito stress test Kit” which actually like the name implies measures the oxidative damage to mitochondria. Moreover, the flowcytometry measurements were performed using Mitsox which is a specific kit for the determination of mitochondria of oxidative damage. Therefore, these two measurements both concluded the level of mitochondria oxidative damage.
We believe that the effect of the ionizing radiation on mitochondria hasn’t been investigated enough compared to ample studies of their effects on the cell's Nucleus. In this work, we chose two different & reliable methods to quantify these effects. Moreover, this line of study is gaining momentum and as evidence to that many very recent articles are targeting this particular point of investigation. [example: Surface Functionalization of Organosilica Nanoparticles With Au Nanoparticles Inhibits Cell Proliferation and Induces Cell Death in 4T1 Mouse Mammary Tumor Cells for DNA and Mitochondrial-Synergized Damage in Radiotherapy by Chihiro Mochizuki published in Frontiers in Chemistry May 2022”. However, the two reviewers believe that the methods presented in this research are insufficient proof for the mitochondria’s damage by IRs and AuNPs. Although, we still think our work has a great value, and the two methods we chose are unique and reliable and the data are obtained from many measurements. We have confidence in our paper because two different approaches resulted in similar outcomes and that confirms our findings and conclusions. We did do the imaging of mitochondria, but we quantified the effect of radiation in another method as imaging itself may not give the required information regarding the effect of the IRs on mitochondria.
Further works are underway for establishing quantitative and certain conclusions levels of these findings. These experiments will focus on another way “3rd” of determination of the levels of the oxidative mitochondria damages and levels of apoptosis to the necrosis enhanced by the gold nanoparticles.
Again, we highly appreciate your efforts in helping making our paper better.
Warm Regards,
Associate professor Moshi Geso